# The Nucleolus: A Central Hub for Ribosome Biogenesis and Cellular Regulatory Signals

**DOI:** 10.3390/ijms26094174

**Published:** 2025-04-28

**Authors:** Donatella Ponti

**Affiliations:** Department of Medical-Surgical Sciences and Biotechnologies, Sapienza University of Rome, Corso Della Repubblica 79, 04100 Latina, Italy; donatella.ponti@uniroma1.it

**Keywords:** nucleolus, cancer, chemoresistance, RNA polymerase I, rDNA, rRNA, UBF, S6K1, ERK, Snail, Myc, rIGSRNA, lncRNA

## Abstract

The nucleolus is the most prominent nuclear domain in eukaryotic cells, primarily responsible for ribosome biogenesis. It synthesizes and processes precursor ribosomal RNA (pre-rRNA) into mature rRNAs, assembling the 40S and 60S ribosomal subunits, which later form the 80S ribosome—the essential molecular machine for protein synthesis. Beyond ribosome production, the nucleolus lacks a delimiting membrane, allowing it to rapidly regulate cellular homeostasis by sequestering key stress response factors. This adaptability enables dynamic changes in size, number, and protein composition in response to cellular stress and signaling. Recent research highlights the nucleolus as a critical regulator of chemoresistance. Given its central role in cell survival and stress adaptation, the nucleolus has become an attractive therapeutic target, particularly in cancer treatment. A deeper understanding of nucleolar metabolism could pave the way for novel therapeutic strategies against various human diseases.

## 1. Introduction

The nucleolus, a structure located within the nucleus of eukaryotic cells, plays a crucial role in producing and assembling ribosomal subunits, which are crucial for protein synthesis [1]. Proliferating cells, including rapidly dividing cancer cells, must achieve sufficient mass and size to support their high protein demand before division. Thus, nucleoli perform a key function in maintenance of homeostasis in cells, and they can directly influence cell cycle progression, cell growth, and proliferation [2]. They synthesize half of the whole transcript pool and utilize up to 80% of energetic and material resources of the cell [3]. These processes involve several hundred protein trans-acting factors and small nucleolar RNAs (snoRNAs), which serve to guide the specificity of ribosomal RNA (rRNA) chemical modifications and ribosomal precursor RNA (pre-rRNA) folding and cleavage. Once ribosomal precursors are released from the nucleolar structure, they undergo further maturation in the nucleolus and nucleoplasm prior to becoming fully functional ribosomal subunits in the cytoplasm, ready to engage in translating mRNAs into proteins. Ribosomal subunits are the structural components of ribosomes, which are responsible for synthesizing proteins in cells. They are composed of two subunits, called the large (60S) and small subunits (40S), which form the functional ribosomes (80S). The large subunit contains three rRNAs, 28S, 5.8S, and 5S, and 50 ribosomal proteins (RPL), while the small subunit contains one rRNA, 18S, and 33 ribosomal proteins (RPS) [4]. rRNAs provide a structural framework for the ribosome, while the proteins help to stabilize the structure and facilitate interactions with other molecules involved in protein synthesis. The transcribed rRNA molecule is used only once, being immediately assembled with ribosomal proteins to produce over 2000–10,000 ribosomes per minute [5].

## 2. Structural and Functional Organization of the Nucleolus

### 2.1. Nucleolus Organizer Regions

The nucleolus forms around nucleolus organizer regions (NORs), which consist of rDNA repeat clusters typically distributed across multiple chromosomes. In humans, NORs are present in the short arm of five acrocentric chromosomes (13, 14, 15, 21, and 22) (Figure 1). The number of repeats varies, ranging between a total of 100–600 copies in a diploid genome. The existence of the nucleolus is determined by the expression of rDNA. Ribosomal DNA represents, with its repetitive sequence, a region particularly prone to DNA damage and rearrangements during repair. The segregation of damaged ribosomal DNA into nucleolar caps has been suggested to prevent inter-chromosomal recombination and promote ribosomal DNA repair [6]. High-resolution microscopy studies have shown that the repression of ribosomal RNA transcription following rDNA double-strand breaks (DSBs) depends on the DNA repair kinases ATM and ATR [7]. Additionally, the nucleolar protein TCOF1 (Treacle) is crucial for nucleolar cap formation in response to rDNA damage. TCOF1 depletion prevents nucleolar cap formation, blocks rDNA transcription silencing, and results in reduced cell viability, increased apoptosis, and genomic instability. Furthermore, TCOF1 facilitates the recruitment of key DNA repair factors, including TOPBP1 and the MRE11-RAD50-NBS1 (MRN) complex, to nucleolar caps [8]. Recent research suggests that the covalent attachment of the ubiquitin-fold modifier 1 (UFM1) protein to target proteins (UFMylation) plays a role in rDNA repair [9,10]. The rDNA unit contains DNA that codes for ribosomal RNA and small nucleolar RNA molecules. The sequences are separated by two internal transcribed spacer (ITS) sequences (ITS1 and ITS2) and flanked by external transcribed spacers (ETS) (3′ ETS and 5′ ETS), which are removed during processing. The fundamental repeat unit consists of a rDNA core that encodes the 47S precursor of rRNA (47S pre-rRNA) and intergenic spacers (IGS). The IGS (IGS1 and IGS2) may contain regulatory elements and other RNAs transcribed by RNA polymerase II [11]. A 400 kb region known as the Distal Junction (DJ), common to all five NOR-bearing chromosomes, is located adjacent to the rDNA arrays and helps anchor them to the nucleolus. In humans, DJs contain a specific insertion of a LINE1 retrotransposon that is unique to primates. This LINE1 element becomes active in human embryonic stem cells (hESCs) and physically interacts with regions of the DJ. Deletion of the LINE1 element disrupts the attachment of the DJ region to the nucleolus, leading to altered nucleolar structure and gene expression, ultimately resulting in the loss of stem cell identity. These findings highlight a key role for the LINE1 element in organizing the genome through the nucleolus in stem cells [12].

### 2.2. Processing of Precursor rRNA

The 47S pre-rRNA is processed into three mature rRNAs, 18S, 5.8S, and 28S rRNA [9]. By contrast, the 5S rRNA gene is transcribed by RNA polymerase III [13]. In humans, the 5S RNA genes are tandemly repeated on chromosome 1 and are localized close to nucleoli. 47S pre-rRNA processing involves a series of specific endo- and exonucleolytic cleavages to remove transcribed sequences that are not part of the mature rRNAs. This process is accompanied by the chemical modification of approximately 200 nucleotides within rRNA sequences. These post-transcriptional modifications (Figure 2) are orchestrated by ribonucleoprotein complexes containing snoRNAs of two types: C/D box snoRNAs and H/ACA box snoRNAs [14]. These snoRNA are associated with fibrillarin, which catalyzes 2′-O-ribose methylation, dyskerin, which catalyzes pseudo-uridylation, and other enzymes involved in rRNA modification [15]. These modifications impact interactions between rRNAs, tRNAs, and mRNAs, and some are known to fine-tune translation rates and efficiency. Ribose 2′-O-methylation is the most abundant rRNA chemical modification and displays a complex pattern in rRNA. These modifications occur early in ribosomal RNA maturation, and although their precise function remains unclear, snoRNAs play a crucial role in directing the modification machinery to specific target sites. These modifications target nucleotides in functionally critical regions of the ribosome, including the peptidyl transferase center, which is essential for peptide bond formation during protein synthesis. The U3 C/D box snoRNP plays a crucial role in 18S rRNA maturation by base-pairing with both the 5′-ETS and 18S rRNA [16]. It coordinates the formation of the 18S rRNA pseudoknot while facilitating early cleavages in the 5′-ETS and ITS1. This essential function makes U3 snoRNA indispensable for 18S rRNA production and, consequently, for cell viability [17]. Like U3, other snoRNPs such as U8, U14, U17, and U22 play a crucial role in multiple pre-rRNA processing steps by chaperoning pre-rRNAs. The folding and processing of pre-rRNA are tightly regulated by both ribosomal proteins (RPs) and ribosomal assembly factors (RAFs). Most ribosomal proteins are essential for ribosomal subunit synthesis, with their incorporation closely coordinated with pre-rRNA folding and RAF recruitment. Indeed, depletion of small-subunit ribosomal proteins blocks pre-rRNA maturation. RAFs transiently associate with forming ribosomal subunits, contributing to enzymatic, structural, and regulatory functions. Synthesis of the 5S rRNA requires a specific regulatory factor called transcription factor IIIA (TFIIIA) [18]. This factor associates with the general class III initiation factors TFIIIB and TFIIIC on the 5S gene promoter and stimulates transcription. The promoter element necessary for 5S rRNA gene transcription is in the transcribed region. After processing the 3′ ends, the 5S rRNA is associated with the ribosomal protein L5 (RPL5). A specialized importin called Syo1 mediates nuclear import of the ribosomal proteins RPL5 and RPL11 and likely chaperones the assembly with the 5S RNA [19]. This complex is then addressed to the nucleolus where it is incorporated into the 60S particles. Also, the incorporation of the 5S ribonuclear protein (RNP) into pre-60S particles depends on association with ribosomal assembly factors. In mammalian cells, the free 5S-RPL5 has ribosome-independent functions in cell cycle regulation, as shown by its capacity to modulate p53 activity. Under nucleolar stress, the 5S RNP in a free pool can sequester the ubiquitin ligase Mdm2, which stabilizes p53 levels and can cause cell cycle arrest and apoptosis [19].

### 2.3. Nucleolar Compartments

Nucleolar function is accompanied by organization of the nucleolus into distinct sub-compartments. Transmission electron microscopy has revealed three major structures within nucleoli: fibrillar centers (FCs), dense fibrillar components (DFCs), and the granular component (GC) [18,19,20]. rDNA transcription units are found in the FC and consist of tandem repeats of these genes. rRNAs are harbored within the DFC and are processed there. Fibrillarin (FBL) and Nucleophosmin (NPM1) are canonical domain markers for the DFC and the GC, respectively. FBL and NPM1 play successive roles in ribosome biogenesis. Later stages of rRNA processing take place in the GC. Thus, the processing of rRNA is spatially arranged in accordance with the ultrastructure of these compartments. One of the most intriguing aspects of nucleolar function is the continuous flux of its components [21]. Significantly, rRNA at various stages of processing moves through the nucleolus, while precursor ribosomal subunits are constantly exported from the nucleoplasm to the cytoplasm for further assembly. This is often described as a vectorial process where the ribosome biogenesis steps such as the precursor rRNA transcription must occur before rRNA modification and binding by ribosomal proteins. Recent studies highlight that the directional flux within the nucleolus may be driven by forces such as those described by the liquid–liquid phase separation model [21]. Proteomic analysis revealed over 500 proteins that localize to the nucleolus. These proteins are involved in cell cycle control, DNA processing, DNA damage response, and repair, in addition to the many proteins connected with ribosome subunit production [22,23]. Ribosome levels can vary across different cell types and stages. A recent study found that fluctuations in ribosome synthesis during the circadian rhythm are linked to liver metabolism. In hepatocytes, protein production peaks at night are followed by equivalent protein degradation during the day. This rhythm is controlled by the target of rapamycin (TOR) pathway, which is activated by nutrients, mainly amino acids. While the impact of circadian and feeding rhythms on transcription is well documented, the rhythmic coordination of mRNA translation and ribosome biogenesis has only recently been discovered [24].

## 3. Regulation of Ribosomal RNA Transcription

The eukaryotic ribosomal rRNA genes 18S, 28S, and 5.8S are transcribed by RNA polymerase I. The recruitment of RNA Pol I to the transcription start site is a result of a series of interactions between specific transcription factors and rDNA promoter. The promoter of the ribosomal gene contains two critical elements: the core element and the upstream control element (UCE). Two transcription factors are required to efficiently activate the rDNA promoter: the selective factor 1 (SL1) and the upstream binding factor (UBF). RNA polymerase I can interact with both the UBF and SL1. Cooperative interaction between these factors creates a stable template, with two UBF molecules forming a dimer bound to the upstream promoter element and at least one SL1 molecule bound to the core promoter element. The UBF binding to promoter elements induces a structural conformation that mimics a nucleosome fold, facilitating the initiation of transcription [25]. In addition, the retinoblastoma protein (pRb) interacts with the UBF and prevents UBF-dependent activation of rDNA transcription. On the contrary, the SV40 large T antigen activates RNA polymerase I transcription by interacting with SL1. Both SL1 and the UBF are subject to regulation via phosphorylation and acetylation [26,27]. The UBF recruits and activates selectivity factor 1 (SL-1) (Figure 3), which consists of a TATA-binding protein (TBP) and five additional factors: TAFI110, TAFI48, TAFI63, TAFI12, and TAFI41 [28,29,30].

### 3.1. RNA Polymerase I Activity

In mammalian cells, RNA polymerase I-activated complexes were found to contain core RNA polymerase I subunits and the RRN3 preinitiation factor. Both genetic and biochemical experiments have demonstrated that RRN3 is essential for rDNA transcription. Current models suggest that RRN3 acts as a bridge between RNA polymerase I and the committed rDNA promoter [31]. RNA polymerase I contains four peripheral subunits unique to this enzyme: A43 (human RPA43), A14, A49, and A34.5. Preinitiation factor RRN3 binds to the A43-A14 stalk and recruits the RRN3-Pol I complex to the rDNA promoter. RRN3 directly associates with the A43 subunit of RNA Polymerase I, enabling the enzyme for transcription initiation [32]. A bioinformatic approach to the sequence of RRN3 identified a domain with similarity identity to the DNA-binding domain of heat shock transcription factor 2 [33]. Similarly, the association of RNA polymerase I-specific transcription initiation factor RRN3 with RPA43 prevents enzyme dimerization and maintains RNA polymerase I in its monomeric form [34]. The TAFI63 and TAFI110 subunits interact with RRN3 on the RRN3-Pol I complex and recruit RNA polymerase I to the rDNA promoter. Together, these proteins form the preinitiation complex (PIC) that binds the promoter favorable for transcription initiation. RNA polymerase I transitions from its open complex, where it is bound to DNA, into its elongation complex, actively synthesizing rRNA. During the elongation, in the active site, two magnesium cations in the catalytic domain coordinate a NTP condensation reaction. Moreover, when RNA polymerase I encounters a DNA lesion or a misincorporated nucleotide, it undergoes a transcriptional pause, awaiting the activation of the cell’s DNA damage repair mechanisms. During the transcription process, the RNA polymerase I can catalyze RNA cleavage. This activity requires the homologous subunits A12.2, Rpb9, and C11. Inefficient RNA cleavage further leads to proofreading errors [35,36]. The transcription termination elements are positioned on two separate sites on the rDNA gene repeat: at the 3′ end of the transcribed region and upstream of the transcription start site. Transcription termination factor I (TTF-I) binds to the termination element at the 3′ ends of the transcribed region and triggers the RNA polymerase I to pause. The subunit A12.2 is required for RNA Pol I release from the DNA template. External factors, such as zinc availability and temperature, also influence RNA polymerase I activity. Conditions that harm cell growth, including stress and nutrient starvation, downregulate transcription of rDNA genes, whereas agents that stimulate growth and proliferation upregulate rDNA transcription. TIF-IA, the RNA Pol I-associated transcription factor that transmits external signals to the nucleolar transcription machinery, is targeted by a variety of protein kinases that phosphorylate TIF-IA at multiple sites. Energy deficits and a high AMP/ATP ratio lead to the phosphorylation and inactivation of RRN3 [37]. Moreover, rDNA transcription fluctuates during the cell cycle, being low in the early G_1_ phase, reaching the highest levels in the S and G_2_ phase, and being shut off in mitosis. Cell cycle-dependent transcription of RNA polymerase I is also achieved by post-translational modifications of the SL-1 and UBF transcription factors. During the S and G2 phases, both SL-1 and the UBF are fully active, contributing to transcription initiation during these stages [38]. During mitosis, SL-1 is phosphorylated, preventing its interaction with the UBF and impairing transcription initiation. SL-1 is dephosphorylated at the end of mitosis, but the UBF remains inactive until late in the G1 phase. In proliferating cells, UBF phosphorylation enhances its interaction with SL-1 and RNA Pol I, whereas in quiescent cells, the UBF remains hypophosphorylated and inactive. Additionally, UBF acetylation, regulated by the CBP acetyltransferase, promotes rDNA transcription by counteracting the repressive effects of tumor suppressors [39]. Notably, the retinoblastoma protein (pRb) has been shown to suppress ribosomal transcription both in vitro and in vivo by directly interacting with the UBF.

### 3.2. RNA Polymerase I and the Proto-Oncogene Myc

Myc is the most powerful inducer of ribosome biogenesis. It stimulates the expression of RNA polymerase I-associated transcription factors and other nucleolar proteins, enabling high-RNA-polymerase-I transcription rates and increased ribosome biogenesis [39,40]. Nucleophosmin (NPM1) is a highly abundant nucleolar protein that interacts with Myc and directs its nucleolar localization. The hyperactivation of nucleoli, which can be triggered by oncogene activation, plays a critical role in carcinogenesis and cancer progression. The transition from cellular quiescence to cell cycle entry and proliferation is driven by the Myc-dependent attachment of ribosomal DNA to the nuclear matrix via the non-transcribed intergenic spacer (IGS) region of rDNA [41]. Oncogenic Myc physically associates with rDNA and remodels the chromatin. Similarly, the Myc inhibitor reduces the induction of pre-rRNA levels as well as the level of matrix attachment in growing cells. The 10058-F4 Myc inhibitor inhibits the interaction of Myc with its hetero-dimerization partner, Max, suggesting the possibility that Myc/Max could bind to E-box sites that are found throughout the IGS sequence [42]. Additionally, genes encoding protein components of the ribosomes displayed an increased mRNA expression upon Myc activation [43]. Myc enhances rRNA upregulation by facilitating the binding of DNA consensus elements to selectivity factor 1 (SL1), which subsequently recruits RNA polymerase I and stimulates rRNA transcription, driving cell cycle entry. Notably, Myc is frequently overexpressed in various cancers [44].

### 3.3. RNA Polymerase I and the Tumor Suppressor p53

In normal cells, surveillance systems based on tumor suppressors have evolved to counteract excessive changes in ribosome biosynthesis and inhibit cell growth. Ribosome biogenesis is regulated by several tumor suppressors, including the Alternative Reading Frame (ARF), p53, phosphatase and tensin homolog (PTEN), and retinoblastoma protein (pRB). The ARF protein has been shown to regulate the cell cycle through both p53-dependent and p53-independent pathways [45]. The ARF is also recognized as a negative regulator of rRNA transcription and maturation. Specifically, the ARF binds to and inhibits the phosphorylation of the upstream binding transcription factor (UBF) [46]. Additionally, the ARF promotes the sumoylation of various interacting proteins, such as Topoisomerase I, MDM2, p53, and the early growth response (EGR1) protein [47,48]. The ARF-mediated sumoylation of EGR1 is essential for PTEN activation, which directly regulates cell size and protein synthesis [49,50]. Tumor suppressors like pRB and p53 further inhibit RNA polymerase I activity and disrupt the assembly of the transcriptional machinery on the rDNA promoter. Under stress conditions, the ARF sequesters MDM2, an E3 ubiquitin ligase, thereby stabilizing p53 levels in the nucleolus. In the nucleolus, p53 inhibits RNA polymerase I activity by sequestering the SL1 factor. Another regulator of RNA polymerase I is the early growth response 1 (EGR1) protein. Activated by stress signals, it functions as a negative regulator of RNA polymerase I. EGR1 localizes to the nucleolus, where its function is closely associated with the expression of the nucleolar proteins nucleophosmin (NPM1) and ARF [49].

## 4. Ribosomal DNA and Epigenetic Modifications

The nucleolus exhibits significant structural diversity during development and across different tissues, along with a dynamic organization that involves the assembly and disassembly of its components throughout the cell cycle, DNA repair, and in response to stress. The number of active rRNA genes at the ribosomal DNA gene repeat is regulated by epigenetic mechanisms. During differentiation, the nucleosome remodeling complex (NoRC) recruits DNA methyltransferases to active rDNA genes and establishes silent heterochromatin at the nucleolus. These silenced rDNA genes are important for genome stability and form hubs around which non-ribosomal DNA organizes into nucleolus-associated domains (NADs). The rDNA of actively transcribed genes exists in an euchromatin configuration that is characterized by DNA hypomethylation, H4ac, and H3K4me2 [51,52]. The rDNA methylation and nucleolar size vary across individuals and are associated with age and longevity [53]. Genomic screens for DNA methylation markers of age across non-rDNA segments identified 353 and 90 CpG sites with the ability to predict chronological age in the human and mouse genomes. Dysfunction of NORs can lead to abnormalities in ribosome production, which can have a variety of consequences for cellular function. Several nucleolar proteins, transcription regulators, and chromatin modulators contribute to determining nucleolus structures [54]. The epigenetic state of rDNA regulates not only ribosome biogenesis, but also the spatial organization and transcriptional activity of the genome. The ribosomal DNA gene is found in three different transcriptional states: inactive, pending, and active [55]. Inactive NORs localize outside the nucleolus, while the pending NOR localizes in the nucleolus and has silenced units that could be easily activated. The active NORs localize in the nucleolus and the epigenetic status of *rDNA* is controlled by multiple factors.

### 4.1. UBF Orchestrates Nucleolar Architecture

The UBF is a multi-HMGB-box protein that acts both as an epigenetic factor to establish the open conformation of chromatin on ribosomal genes and as a basal transcription factor in RNA polymerase I transcription [56,57,58]. It plays an essential role in the aggregation of nucleolar proteins resulting in nucleosome-like structures. Active rDNA genes are nucleosome-free and bound by the UBF and transcription intermediary factor B (TIF-1B), which initiate transcription by RNA polymerase I. In contrast, the promoters of silent rRNA genes are methylated and their coding regions are packed by nucleosomes with repressive histone marks. In addition to binding at the rDNA promoter, the UBF also binds throughout the rDNA coding region and the IGS sequence. It displaces linker histone H1 and contributes to the decondensed state of the euchromatic rDNA [59,60]. The rDNA of silent genes exists in a closed heterochromatin state, characterized by H3K9me, H3K20me, and CpG methylation. Approximately half of the rRNA genes are maintained in an active state [61].

### 4.2. rRNA Transcription and Transcription Termination Factor 1

A key regulator of rRNA transcription is transcription termination factor 1 (TTF-I), a multifunctional nucleolar protein that binds to terminator elements downstream of rDNA [62,63]. Structurally, TTF-I consists of three main domains: a C-terminal DNA-binding domain, essential for recognizing terminator elements, a central domain, required for transcription termination, transcriptional activation, and replication fork arrest, and a N-terminal negative regulatory domain (NRD), which inhibits DNA binding. Interactions between the NRD and the C-terminal domain mask the DNA-binding domain, thereby modulating TTF-I activity. TTF-1 recognizes a consensus sequence known as the Sal box, an 11-base pair motif that can be repeated up to 10 times downstream of the 3′ end of the pre-rRNA sequence. The binding of TTF-I to the Sal box is crucial for halting RNA polymerase I elongation, thereby facilitating pre-rRNA synthesis termination through the formation of a replication fork arrest. TTF-I plays a dual role in rDNA regulation: it terminates ribosomal gene transcription and mediates replication fork arrest but also regulates RNA polymerase I transcription. In addition to these terminator elements, another TTF-I binding site is typically located 170 base pairs upstream of the transcription start site, where it plays a crucial role in transcriptional regulation. It can trigger nucleosome remodeling and antagonize repression of ribosomal gene transcription on chromatin templates. TTF1 interacts with two critical components, the cockayne syndrome B protein (CSB) and the nucleolar remodeling complex (NoRC). NoRC has been shown to play an essential role in rDNA silencing. However, interaction with TIP5, a subunit of NoRC, recovers DNA-binding activity and facilitates both DNA methylation and histone deacetylation, resulting in the silencing of rDNA [64,65]. Conversely, when TTF1 binds to CSB, it activates chromatin through remodeling and epigenetic modifications, thereby promoting ribosomal gene transcription. The activity of TTF1 is further regulated by nucleolin (C23), which prevents the recruitment of TIP5 and histone deacetylase 1 (HDAC1), key factors in establishing a repressive heterochromatin state. Nucleolin depletion results in increased heterochromatin marks (H3K9me2) and reduced euchromatin marks (H4K12Ac and H3K4me3), underscoring its role in maintaining an active chromatin state.

## 5. Stress-Induced Transcription of Noncoding RNA from the Ribosomal Intergenic Spacer

The nucleolus functions as a central stress sensor, detecting cellular imbalances and coordinating adaptive or apoptotic responses. Its ability to monitor and respond to stress positions the nucleolus as a critical regulator of cellular homeostasis and survival. In response to heat shock and acidosis, nucleoli undergo reorganization into reversible amyloid bodies (A-bodies), which form around the intergenic spacer regions of ribosomal RNA [66]. A-bodies are electron-dense, fibrillar structures that contain immobilized proteins in an amyloid-like state.

### 5.1. Long Noncoding RNA (lncRNA)

A recently discovered epigenetic pathway attenuates pre-rRNA synthesis in growth-factor-deprived or density-arrested cells, orchestrated by a long noncoding RNA (lncRNA) that is transcribed by RNA polymerase II from several rDNA genes in an antisense orientation [67]. These transcripts are called PAPAS (“promoter and pre-rRNA antisense”). These lncRNAs localize to the ribosomal intergenic spacer (rIGS), which acts as a hub for a network of regulatory noncoding RNAs. They modulate cellular dynamics by capturing and immobilizing specific proteins within nuclear foci, thereby influencing nucleolar function and stress responses [68,69]. These transcripts lack a common promoter and vary in length from 12 to 16 kb. Their levels increase during cellular quiescence, heat shock, and hypo-osmotic stress. PAPAS interacts directly with rDNA, forming a DNA–RNA triplex structure that tethers PAPAS to a stretch of purines within the enhancer region, thereby guiding the associated nucleosome remodeling and deacetylation complex (CHD4/NuRD) to the rDNA promoter. Under these conditions, the lncRNA transcripts recruit the histone methyltransferase Suv4-20h2 to the rDNA locus, catalyzing the trimethylation of histone H4 at lysine 20 (H4K20me3), a repressive chromatin mark that rapidly suppresses rDNA transcription [70]. In response to heat shock and hypo-osmotic stress, they recruit the Nucleosome Remodeling Deacetylase (NuRD) complex to the rDNA promoter, promoting histone deacetylation and nucleosome repositioning, which further silences rDNA transcription [71].

### 5.2. Ribosomal Intergenic Spacer (rIGS RNAs)

Additionally, heat shock induces the expression of two IGS-derived RNAs, IGS RNA 16 and IGS RNA 22, transcribed by RNA polymerase I from regions located 16 kb and 22 kb downstream of the pre-rRNA transcription start site, respectively [72,73]. Together with IGS RNA 28, these mature IGS transcripts bind proteins at their expression sites within the rDNA cassette. An alternative IGS RNA is pRNA that is transcribed from spacer promoters upstream of the pre-rRNA transcription start site. Although the exact mechanism by which IGS RNAs interact with rDNA and associated proteins is unclear, both RNA–DNA triplexes and R-loops have been proposed. Triplex structures involve base pairing between the lncRNA and the major groove of the DNA duplex [74]. R-loops are three-stranded structures composed of an RNA–DNA hybrid and a displaced single-stranded DNA strand, relying on canonical base pairing. Together, these findings suggest that lncRNA-dependent changes in nucleolar architecture may serve as a general mechanism to safeguard cellular homeostasis [71].

## 6. RNA Polymerase I and Cell Signaling

The cellular response to stress is a crucial adaptation mechanism to environmental changes. This response is characterized by significant alterations in gene expression, enabling the cell to maintain homeostasis and survival. Nucleolar metabolism is influenced by the interaction between pathways activated from extracellular signals to coordinate ribosome synthesis and cell proliferation [75].

### 6.1. RNA Polymerase I and mTOR Signaling

The regulation of RNA polymerase I is governed by post-translational modifications in response to external stimuli. One of the key regulatory players in this process is the mammalian target of rapamycin (mTOR) protein. The mTOR signaling pathway integrates a wide range of signals to control genes involved in the control of cellular growth and nutrient response. A major effect of mTOR signaling is the upregulation of RNA polymerase I transcription, mediated through the mTOR complex 1 (mTORC1) and its downstream activation of the kinase S6K1 in response to increased nutrient availability [76]. S6K1 activation leads to the phosphorylation of the initiation factor RRN3 and the UBF [76], promoting RNA polymerase I activity (Figure 4). At the same time, the increased nucleotide demand driven by increased RNA Pol I activity is achieved by mTORC1-dependent stimulation of purine and pyrimidine synthesis pathways. During increased cell growth, mTOR and protein kinase 2 (CK2) further phosphorylate the UBF, thereby increasing RNA polymerase I activity [77].

### 6.2. RNA Polymerase I Activity and MAPK Signaling

RNA Polymerase I activity can also be enhanced by growth stimuli transmitted via the MAPK signaling pathway. In response to growth factor signaling, Ras-GTP activates the MAPK pathway, which triggers the nuclear import of ERK. Subsequently, ERK phosphorylates and activates RRN3, thereby enhancing RNA Pol I activity [78]. Growth factors that activate the ERK pathway promote UBF phosphorylation, enhancing its interaction with rDNA and increasing RNA Pol I transcription [79]. However, the level of 47S pre-rRNA decreases to varying extents depending on specific stress-inducing agents. Stress conditions activate the p38-MAPK and c-Jun N-terminal kinase (JNK) pathways, which collaborate via shared upstream and downstream effectors to regulate gene expression in response to environmental challenges. This coordinated response, known as the environmental stress response (ESR), is triggered by factors such as osmotic and oxidative stress, inflammatory cytokines, and ribotoxic agents. Notably, ribosomal rRNA synthesis is more susceptible to oxidative stress than overall mRNA synthesis. JNKs, classified as stress-activated protein kinases (SAPKs), play a pivotal role in the cellular response to environmental stress, balancing pro-survival signals with pro-apoptotic pathways. Their activity is induced by alkylating agents, hyperosmotic shock, proinflammatory cytokines, and oxidative stress. Under such conditions, RNA polymerase I activity is impaired due to TIF-IA inactivation, mediated by JNK-dependent phosphorylation at threonine 200. This modification disrupts TIF-IA’s interaction with RNA Pol I and TIF-IB/SL1, preventing transcription initiation complex formation and causing TIF-IA translocation from the nucleolus to the nucleoplasm. Various stress stimuli also induce nucleolar protein relocalization. Tumor suppressor proteins like p53 are stabilized by nucleolar sequestration, while nucleolin (C23) and nucleophosmin (NPM1) are translocated to the nucleoplasm, where they contribute to cell cycle arrest [80]. Additionally, under nutrient deprivation, cells downregulate energy-intensive processes, such as ribosome biogenesis and protein synthesis. Low ATP levels suppress rDNA transcription, whereas elevated ATP levels enhance it, reflecting the tight coupling of ribosomal RNA synthesis with cellular energy availability. AMP-activated protein kinase (AMPK) plays a crucial role in translating changes in energy levels into adaptive cellular responses. The activation of AMPK by glucose deprivation or treatment with the AMP-mimetic drug AICAR (5-amino-4-imidazolecarboxamide ribonucleotide) leads to inactivation of TIF-IA both in vivo and in vitro. AMPK phosphorylates TIF-IA at serine 635 and this phosphorylation impairs the interaction of TIF-IA with the TBP-containing promoter selectivity factor SL1. Consequently, recruitment of RNA Pol I to the rDNA promoter and transcription complex formation is impaired.

### 6.3. ErbB Receptors in the Nucleolus

Recently, members of the EGFR family have also been identified in the nucleolus of both normal and cancer cells [81]. These tyrosine kinase receptors have a large glycosylated extracellular domain, a hydrophobic transmembrane region, and a cytoplasmic domain, which carry tyrosine kinase activity [82]. ErbB receptors are expressed in various tissues of epithelial, mesenchymal, and neuronal origin. They have been implicated in various aspects of neural development, including circuit generation, axon lining, neurotransmission, and synaptic plasticity. Substantial evidence supports the involvement of the ErbB family in the development and progression of various types of cancer [83]. In numerous cells, multiple members of the ErbB receptor family are co-expressed. Generally, these receptors function on the cell membrane, where they bind to growth factors or specific ligands. Upon binding, they undergo homo- or heterodimerization, activating their intrinsic tyrosine kinase activity and recruiting adaptor proteins that trigger downstream signaling pathways [84]. This signaling generates crosstalk with multiple cellular processes, such as proliferation and cell motility. Additionally, mitogenic growth factors, including neuregulin-1 and its isoform NRG1-β3, and ligands of the ErbB3 receptor have been identified in the nucleus and nucleolus of the cell. ErbB1 and ErbB2 receptors have been identified in the nucleus as full-length proteins, while ErbB3 and ErbB4 have been found in the nucleus and nucleolus as truncated isoforms [85,86,87]. Two shorter ErbB3 variants have been identified in the nucleus and nucleolus of human cells. The first variant, an 80 kDa ErbB3, is involved in regulating the transcription of the cyclin D1 gene [81,88]. The second variant, a 50 kDa ErbB3, colocalizes with fibrillarin in the nucleoli of various tumor cell lines and primary glioblastoma cells [89] and relocates from the nucleolus to the cytoplasm, when the levels of the NRG-1 ligand increase, promoting pre-rRNA synthesis. The 50 kDa variant of ErbB3 found in glioblastoma nucleoli also interacts with key ribosome biogenesis factors, such as the upstream binding factor (UBF). Silencing ErbB3 in glioma cell lines blocks the cell cycle and reduces proliferation. This effect is disrupted by actinomycin D, which causes the nuclear accumulation of ErbB3 in association with the nucleolar protein nucleolin (C23). Nucleolin is an abundantly expressed nucleolar phosphoprotein of exponentially growing cells. It is present in abundance at the dense fibrillar and granular regions of the nucleolus. This nucleolar protein is involved in the control of the transcription of ribosomal RNA (rRNA), ribosome assembly, and nucleocytoplasmic delivery of ribosomal subunits. Recently, it was demonstrated that nucleolin is also present on the surface of several cells, and its overexpression has been correlated with tumor grade, suggesting a crucial role in tumor progression [90]. Despite these findings, the nuclear functions of ErbBs remain unclear, warranting further investigation into its impact on ribosome biogenesis and tumor progression.

## 7. The Nucleolus in Cancer: A Hidden Driver

The nucleolus plays a crucial role as a sensor of various cellular stresses, including genotoxic and oxidative stress, nutrient deprivation, and oncogene activation [91]. Nucleolar morphology, including its shape, size, and number per nucleus, can be profoundly remodeled in disease. A well-established positive correlation exists between nucleolar size and the intensity of ribosome biosynthesis, particularly in cells undergoing neoplastic transformation and during cancer progression [92,93]. Notably, dysregulated synthesis of pre-rRNA transcripts has been associated with poor cancer prognosis [94,95]. Additionally, prominent nucleoli have frequently been linked to worse cancer outcomes [96,97,98]. In various tumors, intense staining of the short arms of chromosomes corresponding to nucleolar organizer regions (NORs) correlates with increased expression of ribosomal RNA genes [99]. Beyond cancer, nucleolar dysfunction has been implicated in a range of human diseases, particularly ribosomopathies. These disorders can be classified into two groups, with one major category being inherited bone marrow failure syndrome (IBMFS) and Treacher Collins syndrome. Notably, IBMFS-related ribosomopathies predispose patients to cancer development [100,101,102]. In a distinct context, mutations in the Nibrin gene (NBS1) impair DNA damage response and genomic stability, leading to Nijmegen breakage syndrome. This syndrome is characterized by progressive microcephaly and a significantly increased risk of cancer [103]. Nibrin has recently been identified as a nucleolar protein. Within the nucleolus, NBS1 localizes to the fibrillar center (FC), where it interacts with the ribosomal RNA (rRNA) transcription machinery, including RNA polymerase I. Its C-terminal domain binds to pre-rRNA and associated processing factors, indicating a direct role in rRNA synthesis and maturation. Nibrin is a crucial component of the MRE11-RAD50-NBN complex (MRN) [104]. It is involved in the response to DNA double-strand breaks (DSBs) and represents an attractive strategy to overcome tumor drug resistance. Lactylation of NBS1 at lysine 388 is critical for the formation of the MRN complex and the recruitment of homologous recombination proteins to sites of DNA double-strand breaks [105].

### 7.1. Fibrillarin and Ribosome Heterogeneity

Enlarged nucleoli and increased cell proliferation are observed along with more intensive rDNA transcriptional activity, which is frequently associated with enhanced expression of factors involved in various stages of ribosome biosynthesis, such as the upstream binding factor (UBF), DNA topoisomerase I, fibrillarin (FBL), nucleolin (C23), and nucleophosmin (NMPI or B23), as well as small nucleolar RNAs (snoRNAs) and ribosomal proteins (RPs). Increased cancer cell aggressiveness has been linked not only to enhanced rRNA biosynthesis but also to the activation of specific pre-rRNA maturation pathways. These pathways introduce new post-transcriptional modifications into rRNA, leading to elevated ribosome production and altered translational functionality [106]. This alternative rRNA maturation process during ribosome biogenesis assigns new regulatory roles to rRNAs within the ribosome. Beyond serving as structural components, rRNAs actively modulate mRNA decoding during translation. Ribosome heterogeneity, driven by changes in rRNA post-transcriptional modifications, ribosomal protein stoichiometry, and post-translational modifications of ribosomal proteins, plays a crucial role in cancer development [107]. These alterations result in specialized ribosomes that selectively enhance the translation of specific proteins, potentially promoting tumor growth and progression [108]. A key nucleolar protein involved in post-transcriptional modifications of ribosomal precursor rRNAs is fibrillarin. As a C/D box snoRNA-guided methyltransferase, fibrillarin plays a critical role in rRNA 2′-O-methylation, pre-rRNA processing, and ribosome assembly. Recent studies have demonstrated that rRNA 2′-O-methylation is dynamic, influencing ribosomal translational capabilities and potentially altering protein synthesis in cancer [17,109]. Fibrillarin not only facilitates rRNA methylation but also regulates RNA polymerase I activity through the methylation of rDNA gene promoters. Specifically, it modulates the methylation of a glutamine residue on +H2A histone at the glutamine 104, thereby influencing RNA Pol I transcription [110]. Fibrillarin expression is highly regulated in various physiological and pathological contexts, including development, stem cell differentiation, viral infections, and cancer. In cancer models, nucleolar protein fibrillarin modulation influences tumor progression, with sustained expression prolonging the pluripotent state of mouse embryonic stem cells [111]. In breast cancer cells, the altered fibrillarin expression correlates with changes in rRNA 2′-O-methylation, affecting translational accuracy and the initiation of mRNAs containing internal ribosome entry site (IRES) elements [112,113]. In recent studies, fibrillarin knockdown has been shown to lead to site-specific alterations in 2′-O-methylation, including at critical ribosomal positions, selectively modifying ribosome function. These findings highlight the extensive modulation potential of rRNA 2′-O-methylation patterns. The site-specific effects of fibrillarin knockdown suggest that regulating common components of the methylation machinery can fine-tune 2′-O-methylation patterns, thereby influencing ribosome function. The observed downregulation of fibrillarin during neurogenesis and stem cell differentiation may directly impact rRNA modification and ribosome activity [114]. Conversely, fibrillarin overexpression in tumors could increase 2′-O-methylation at specific sites, altering translational control in cancer cells.

### 7.2. Nucleolus and Epithelial–Mesenchymal Transition (EMT)

The induction of rDNA transcription is also associated with cellular plasticity, dedifferentiation, and stemness. The nucleolus has been identified as a localization site for the transcriptional factor Snail, a key inducer of epithelial–mesenchymal transition (EMT). EMT is a crucial process during embryonic development. During this process, epithelial cells acquire fibroblast-like characteristics, exhibit decreased intercellular adhesion and display increased cell motility [115,116]. Additionally, Snail contributes to tumor progression by downregulating EMT-associated genes, including E-cadherin (E-cad) and claudin. E-cadherin is a member of the cadherin family, a component of adhesion junctions, and the primary organizer of the epithelial phenotype [117,118,119]. The loss of E-cad expression is associated with tumor metastasis, and induced expression of E-cad in cancer cells can prevent tumor progression and invasion [120,121,122]. Additionally, increased Snail levels contribute to tumor resistance against various chemotherapeutic drugs [123]. Snail activity also induces the expression of genes associated with an invasive phenotype, such as fibronectin and the metalloprotease 9 (MMP9).

#### 7.2.1. Snail and Signaling Network

The expression of Snail is regulated by a complex signaling network that includes integrin-linked kinase (ILK), phosphatidylinositol 3-kinase (PI3K), mitogen-activated protein kinases (MAPKs), glycogen synthase kinase 3-beta (GSK-3β), and NF-κB. The TGF-β/Smad pathway, which induces EMT in hepatocytes, epithelial cells, and mesothelial cells, activates Snail expression. Additionally, Notch signaling employs distinct mechanisms that act synergistically to regulate Snail synthesis. Snail’s subcellular localization is regulated by phosphorylation, particularly through p21-activated kinase 1 (PAK1). This enzyme phosphorylates Snail at serine 246, promoting its nuclear localization and enhancing its transcriptional activity. The stable and active form of Snail localizes only in the nucleus of the cell. In the cytoplasm, Snail has a short half-life due to ubiquitin-mediated proteasomal degradation. Several E3 ubiquitin ligases, including FBXW7 and FBXL14, promote Snail degradation via the proteasome [124]. Conversely, deubiquitinates enzyme such as USP3 and USP11 play a crucial role in regulating Snail stability, removing the ubiquitin subunits from the protein.

#### 7.2.2. USP36-Snail Nucleolar Axis

Nucleolar accumulation of Snail is induced specifically by ribotoxic stress, characterized by the activation of the nucleolar deubiquitinase USP36, which consequently increases Snail1 stability. The nucleolar localization of Snail is positively correlated with an increase in 47S pre-rRNA levels, and the silencing of USP36 abrogates this effect [123]. Conversely, the Snail K157R mutant is unable to accumulate in the nucleolus, and Snail knockdown results in a significant reduction in 47S pre-rRNA expression. Furthermore, the USP36 knockdown also strongly inhibits 47S pre-rRNA expression. All these findings have addressed the identification of a new USP36/Snail nucleolar axis that promotes ribosome biogenesis. The increase in the expression of ribosomal precursor 47S pre-rRNA induced by Snail improves ribosome biogenesis and support cancer cell survival under stress conditions. Also, chemotherapeutic agents such as doxorubicin, translation inhibitors like cycloheximide, and well-known ribotoxic stress inducers such as puromycin and blasticidin disrupt ribosome function by activating p38/JNK signaling pathways, leading to an increase in nucleolar USP36 and Snail1 protein expression. In contrast, rapamycin, an mTOR inhibitor, fails to activate JNK signaling or upregulate USP36 and Snail.

#### 7.2.3. RPL11-c-Myc-Snail Axis

The nucleolar USP36–Snail axis includes another crucial player component, ribosomal regulatory protein 1 (RRS1) [125]. RRS1 regulates ribosome biogenesis by recruiting the 5S ribonucleoprotein (RNP) to form the pre-60S ribosomal subunit with ribosomal production factor 2 (Rpf2). Its depletion results in the accumulation of the pre-60 subunit in the nucleoplasm and the stalling of ribosome synthesis. RRS1 is also a component of a second important axis, the RPL11-c-Myc-Snail axis. This axis is involved in the regulation of the invasion and metastasis of cancer cells. The accumulation of ribosomal protein RPL11 in the nucleoplasm inhibits c-Myc-dependent transcription of EMT-related genes. USP36 is also a crucial regulator of Myc [126]. It directly binds to Myc and interacts specifically with tumor suppressor Fbw7γ in the nucleolus, but not with Fbw7α present in the nucleoplasm [127]. The downregulation of USP36 significantly reduces the levels of Myc and inhibits cell proliferation. Furthermore, the knockdown of USP36 abolishes c-Myc induction following serum stimulation, demonstrating that USP36 plays a critical role in c-Myc stabilization in response to growth signals. USP36 overexpression has also been observed in many human cancers, such as breast and lung cancers, implying its oncogenic nature.

## 8. Conclusions

The nucleolus has long been considered a distinct subnuclear compartment exclusively dedicated to ribosome biogenesis. Its spatial segregation from key receptors and extracellular signaling pathways has contributed to an underestimation of its function as cell sensor. However, emerging evidence indicates that the nucleolus acts as a dynamic integrative hub where crucial signaling pathways converge with the aim of regulating cellular proliferation and survival. Beyond its primary role in ribosome production, the nucleolus constantly modulates the rate of ribosome biogenesis in response to cellular metabolic demands. The nucleolar sequestration of proto-oncogenes, such as Myc, and tumor suppressors, including p53, plays a pivotal role in the control of ribosome biogenesis, protein synthesis, and cell cycle progression. In cancer, this sophisticated control is deregulated with the scope to induce abnormal synthesis of the ribosome to support the high level of cell proliferation. However, under specific cell growth conditions, the nucleolus can influence differential post-transcriptional modifications of 47S pre-rRNA with the production of ribosome heterogeneity. Recent evidence attributes the role of a critical chemoresistance regulator to the nucleolus. Under ribosome stress conditions of growth, it can counteract the repression of 47S pre-rRNA synthesis, recruiting the Snail transcription factor, a key driver of epithelial–mesenchymal transition (EMT), and inducing cell survival. All these findings draw attention to the new function of the nucleolus as a central player in tumor biology and cellular adaptation. Targeting nucleolar metabolism may provide novel therapeutic strategies by unveiling previously unrecognized molecular mechanisms that play a crucial, deciding role in cell fate.

## Figures and Tables

**Figure 1 ijms-26-04174-f001:**
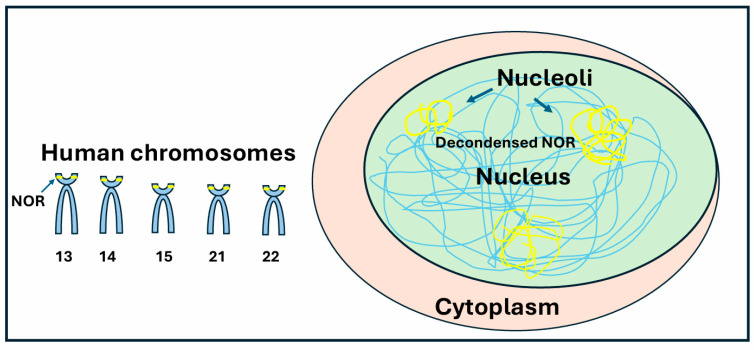
**The nucleolus**. In human cells, nucleoli are localized on specific pairs of chromosomes: 13, 14, 15, 21, and 22. They correspond to the decondensed regions of the nucleolar organizer regions (NORs). However, not all NORs are active in cells. The number of active NORs is positively correlated with pre-rRNA synthesis and protein translation.

**Figure 2 ijms-26-04174-f002:**
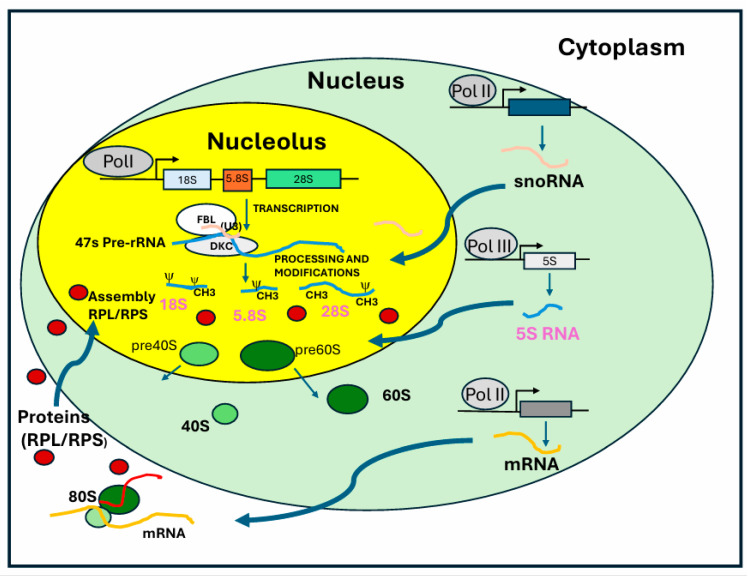
**Ribosome biogenesis.** Ribosome biogenesis takes place in the nucleolus. After rDNA transcription, pre-rRNA undergoes modifications such as methylation and pseudouridylation, catalyzed by the enzymes fibrillarin and dyskerin, respectively. Their catalysis is supported by small nucleolar RNAs (snoRNAs), such as U3. After modification, the rRNA precursor is processed by endonucleases, producing mature 18S, 5.8S, and 28S rRNA. The 5S rRNA is transcribed by RNA Polymerase III. The mature rRNAs interact with ribosomal proteins (RPS and RPL) to form the 40S and 60S ribosomal subunits, respectively. In the cytoplasm, these two subunits assemble with mRNA to form the active 80S ribosome, which is ready for protein translation. All RNA polymerase enzymes (Pol I, Pol II, and Pol III) are tightly coordinated during ribosome biogenesis.

**Figure 3 ijms-26-04174-f003:**
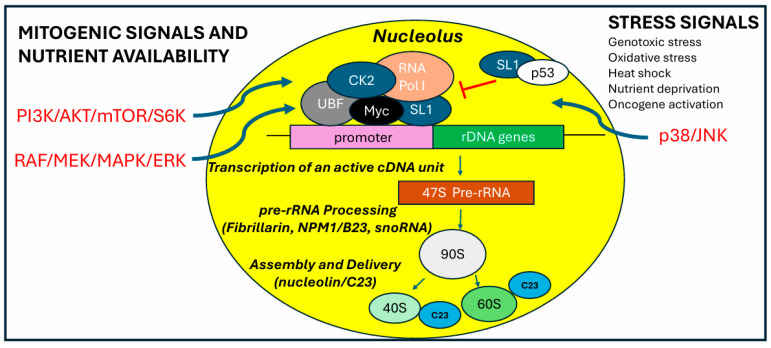
**The mitogenic pathways modulate RNA Polymerase I activity.** The nucleolus functions as a critical site for integrating both positive and negative regulatory signals to coordinate ribosome biogenesis with cell proliferation. RNA Polymerase I activity is regulated by specific cellular pathways such as RAF/MEK/MAPK/ERK and PI3K/AKT/mTOR/S6K. Oncogenes and tumor suppressor proteins play a crucial role in ribosome biogenesis. Under proliferative conditions, the proto-oncogene Myc enhances ribosome biogenesis by positively interacting with the RNA Polymerase I complex. Conversely, under stress conditions, the tumor suppressor p53 sequesters SL1 from the RNA Polymerase I complex, reducing the rate of rRNA transcription.

**Figure 4 ijms-26-04174-f004:**
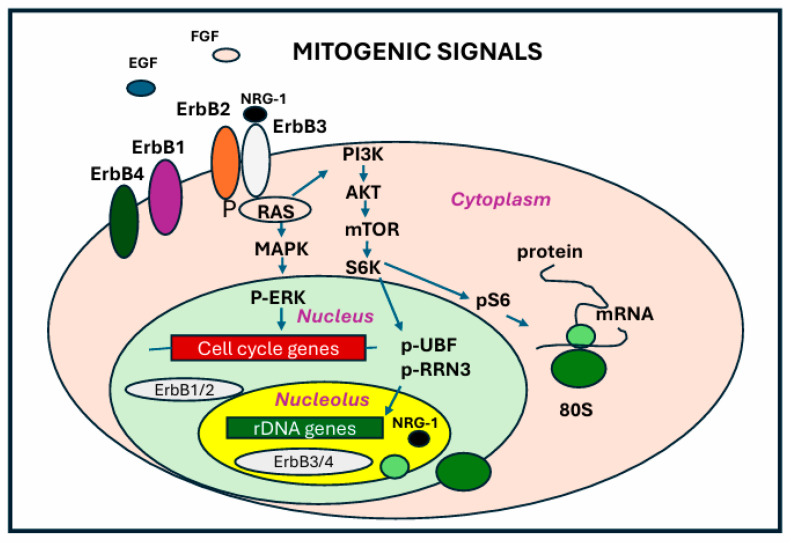
**ErbB receptors and nucleolar metabolism.** The mitogenic pathways activated by specific growth factors and ligands play a crucial role in regulating nucleolar metabolism and ribosome biogenesis. Upon binding to their specific ligands, tyrosine kinase receptors (ErbBs) initiate signaling cascades that activate key regulators of RNA Polymerase I activity, such as UBF. Ribosome biogenesis is highly coordinated with protein synthesis.

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
