# Peer review of "The Nucleolus: A Central Hub for Ribosome Biogenesis and Cellular Regulatory Signals"

_ijms, 2025, doi:10.3390/ijms26094174_

Round 1
Reviewer 1 Report
Comments and Suggestions for Authors
In this manuscript, the author review a good part of the current literature about the nucleolus. In general the review is well written, and cover several aspects of the biology of the nucleolus. I found few typos and maybe some parts should be placed in different order but in general it is a good review that could be useful for the readers and for students. I highligh here few minor comments and some references that I believe could be included in the manuscript.
Line 11: precursor ribosomal RNA is pre-rRNA not rRNA
Line 54: in a diploid genome a full stop is missing
Line 54: represent a s is missing
Line 55: “The segregation of damaged ribosomal DNA into nucleolar caps has been suggested to serve as prevention of inter-chromosomal recombination and to promote ribosomal DNA repair.” Reference is missing
Lines 57-64: “High- resolution microscopy studies...and the MRE11-RAD50-NBS1 (MRN) complex, to nucleolar caps" references are missing
Line 58: “...repression of RNA polymerase I...” it should be first explained which polymerase transcribe rRNA genes
Line 68 “In humans, NORs are present in the short arm of five acrocentric chromosomes (13, 14, 15, 21 and 22)” I would move this sentence to Line 53
Lines 80-93: references are missing
Line 103 reference is missing
Line 185 “Approximately half of the rRNA genes are maintained in an active state.” reference is missing
Line 287 “Nucleophosmin (NPM1 or B23), IS??? a highly”
I would suggest to the Author to add also the findings in these manuscripts:
10.1073/pnas.2422029122
it has been recently showed that NBS1 localizes in the fibrillar center (FC) of the nucleolus and promotes RNA polymerase I-mediated ribosomal RNA (rRNA) biogenesis. It would be important to add also this aspect in the review
10.1101/gad.351979.124
This paper highlights the role of transposable elements in role in nucleolar organization. It would be also interesting to include this aspect in the review.
https://doi.org/10.3390/biom14101333
This review discusses long non-coding RNAs transcribed from the intergenic spacer region of ribosomal DNA (IGS rDNA) and their role in stress. The examples reported in this review could also be discussed in the manuscript
Author Response
Reply to Reviewer 1
I thank the reviewer for their appropriate and constructive review of the manuscript. All suggestions have been carefully considered. The previously missing references have been included, and the suggested ones are now discussed in the appropriate sections. Additionally, some information has been added to the legend of the Graphical Abstract. All the modifications introduced in the text are now highlighted in yellow.
Line 11 – Correction of terminology:
According to the reviewer the term "pre-rRNA" has been introduced
Line 54 – Missing punctuation:
As suggested a full stop has been added to complete the sentence (now line 58).
Line 54 – s is missing
The s has been added (now line 59).
Line 55 – Missing reference:
The appropriate reference for the sentence discussing the segregation of damaged ribosomal DNA into nucleolar caps has been added (now line 62).
Lines 57–64 – Missing references:
The missing references supporting high-resolution microscopy studies and the role of the MRN complex have been introduced as suggested (line 64).
Line 58 – Clarification regarding RNA polymerase I:
I apologize for the inappropriate placement of RNA polymerase I. It has now been replaced with the more general term rRNA transcription.
Line 68 – Sentence repositioning:
According to the reviewer “The sentence describing the location of NORs in acrocentric chromosomes” has been moved to line 56 for improved coherence.
Lines 80–93 – Missing references:
The missing references have been added to support the relevant content (they are now included in line 96-110).
Line 103 – Missing reference:
A reference has been added to support the statement (it is now included in line 118).
Line 185 – Missing reference:
The appropriate reference has been added to the statement regarding the activity of rRNA genes (it is now included in line 317).
Line 287 – Sentence review:
According to the reviewer, the sentence regarding Nucleophosmin (NPM1 or B23) has now been revised. (line 238)
Suggested additional literature:
I thank the reviewer for suggesting insightful papers related to a novel aspect of nucleolar metabolism. All three suggested papers have now been incorporated and discussed in the relevant sections of the manuscript
PNAS 2024 (10.1073/pnas.2422029122): The role of NBS1 in the fibrillar center of the nucleolus and its impact on rRNA biogenesis has been included (lines 512–524).
Genes & Development 2024 (10.1101/gad.351979.124): The involvement of transposable elements in nucleolar organization has been addressed (lines 78–86).
Biomolecules 2024 (https://doi.org/10.3390/biom14101333): The discussion now includes the role of long non-coding RNAs transcribed from the IGS region and their function under stress conditions (lines 350–396).
Reviewer 2 Report
Comments and Suggestions for Authors
This is a timely, well written and comprehensive overview of current insights and state of the art knowledge concerning the structural and functional role of the eukaryotic (mainly human) nucleolus and nucleolar metabolism, towards novel strategies against various human diseases.
One minor concern is, that given the large amount of (relevant) information, I would advise the author to make a more structured use of sub-headings, guiding the interested reader through current knowledge and perspectives. Sub-headings have been used in a few instances, however, not clearly constructively whereas this would certainly be possible for other parts/sections as well.
Author Response
Replay to Reviewer 2
I would like to thank the reviewer for the positive comments and valuable suggestion to improve the organization of the text. To enhance readability and better guide the reader through the extensive information, we have restructured several sections and added additional sub-headings. All the modifications introduced in the text are now highlighted in yellow.